# Availability, cost and affordability of essential cardiovascular disease medicines in the south west region of Cameroon: Preliminary findings from the Cameroon science for disease study

**Anastase Dzudie[1,2,3,4], Epie Njume[2,5]\*, Martin Abanda[2], Leopold Aminde[2,6], Ba Hamadou[3], Bonaventure Dzekem[2], Marcel Azabji[3], Marie Solange Doualla[1,3†], Marcelin Ngowe[1], Andre P. Kengne[2,7]**

1 Faculty of Health Sciences, University of Buea, Buea, Cameroon, 2 Clinical Research Education, Networking & Consultancy (CRENC), Douala, Cameroon, 3 Faculty of Medicine and Biomedical Sciences, University of Yaounde 1, Yaounde, Cameroon, 4 Department of Medicine, Faculty of Health Sciences, University of Cape Town, Cape Town, South Africa, 5 Cameroon Baptist Convention Health Services, Nkwen-Bamenda, Cameroon, 6 School of Public Health, Faculty of Medicine, University of Queensland, Brisbane, Australia, 7 Non-communicable Disease Unit, Medical Research Council, Cape Town, South Africa

† Deceased.
\* enjume20181@outlook.com

## Abstract

### Background

More than 80% of premature deaths due to cardiovascular disease (CVD) occur in low- and middle-income countries. However, access to, and affordability of medications remain a challenge in these countries.

### Objective

To assess the availability, cost and affordability of essential cardiovascular medicines in the South West region of Cameroon.

### Methods

In an audit of 63 medicine outlets, twenty-six essential medicines were surveyed using the World Health Organisation (WHO) /Health Action International methodology. Availability, costs and the ratio of the median price to the international reference price were evaluated in public, confessional, private facility medicine outlets, and community pharmacies. Affordability was assessed by calculating the number of days' wages it will cost the lowest-paid unskilled government worker to purchase a month worth of chronic treatment.

### Findings

Availability ranged from 25.3% (public facility outlets) to 49.2% (community pharmacies) for all medicines. This was higher in urban and semi-urban compared to rural outlets. Cost of

**Data Availability Statement:** Data are available through the server of the South African Medical

Research Council via this link: http://medat.samrc.ac.za/index.php/catalog/38.

**Funding:** The authors received no specific funding for this work.

**Competing interests:** The authors have declared that no competing interests exist.

medicines was highest in community pharmacies and lowest in public facility outlets. Aspirin, digoxin, furosemide, hydrochlorothiazide and nifedipine were affordable (cost a day's wage or less). Medicines for heart failure and dyslipidaemia (beta blockers, angiotensin converting enzyme inhibitors and statins) required 2–5 days and 6–13 days wages respectively for one month of chronic treatment.

## Conclusion

Overall availability of CVD essential medicines was lower than WHO recommendations, and medicines were largely unaffordable. While primary prevention is pivotal, improving availability and affordability of medicines especially for public facilities would provide additional benefit in curbing the CVD burden.

## Introduction

Globally, cardiovascular disease (CVD) is the leading cause of death with 80% of these deaths occurring in low- and middle-income countries (LMIC) [1–4],. In SSA countries as reported elsewhere, the rise in this burden of CVD is influenced by population growth, ageing, increasing urbanisation and lifestyle changes [2, 5, 6]. The cost of CVD on individuals and households, healthcare systems and countries is prohibitive, especially in SSA. This cost is both direct (hospitalisation, rehabilitation services, physician visits, drugs) and indirect: cost associated with morbidity and mortality (losses of productivity due to premature death and short- or long-term disability) [7]. It has been shown that the combined impact of diabetes and CVDs may undermine the future development and prosperity of this region [8–11].

In 2012, the world health organization (WHO) formulated the $25 \times 25$ Goal which aims at achieving a 25% reduction in the number of premature deaths (occurring before 70 years of age) due to non-communicable disease (NCD) by 2025 [12]. This was followed by a Global Action Plan (GAP) that includes, amongst others, a target of 80% availability and affordability of essential medicines (EMs) for treatment and secondary prevention of CVDs and other NCDs, and at least 50% of eligible people to receive drug therapy and counselling (including glycaemic control) to prevent heart attacks and strokes [13, 14]. An availability of medicines for cardiovascular diseases ranging from 9.1% to 50%, and 36.4 to 59.1% in rural and urban settings respectively was reported in the West region of Cameroon in 2012 [15].

There is ample evidence on the use of pharmacological treatment for secondary prevention of cardiovascular events [12, 16, 17], thus affordable access to EM is urgently warranted in efforts to reduce the burden of CVD [18]. It has been estimated that up to 80% of the burden of NCDs in many countries can be reduced with the use of appropriate medicines [14]. In spite of this proven benefit of EM, their availability in many low- and middle-income countries is sub-optimal. The Prospective Urban Rural Epidemiology (PURE) study showed that the use of proven medicines (antiplatelet, β blockers, angiotensin-converting enzyme inhibitors or angiotensin receptor blockers and statins) in patients with coronary heart disease or stroke is low, particularly in low-income countries, where 80% took no medicines [19]. This implies that there is a huge gap between standard clinical practice (based on high-quality evidence) and the current practice in those settings. To bridge this gap, it is vital to understand the potentially modifiable health system barriers at each level of care, in order to inform appropriate solutions.

The aim of this study was thus to assess the availability and affordability of essential CVD medicines, how these vary across various medicine outlets and finally compare the availability and affordability pattern with medications for selected endemic infectious diseases in Cameroon.

## Materials and methods

### Ethics approval and consent to participate

Ethical approval was obtained from the Institutional Ethics Committee for Research on Human Health of the University of Douala (ref: IEC-UD/506/02/2016/T). Administrative approval was obtained the South West Regional Delegation of Public Health. Administrative approval was also obtained from the District Medical Officers where possible, otherwise consent was obtained directly from the head of selected medicine outlets. The purpose of the study was explained to the head of each participating medicine outlet and to any other informant. Participants were not under any obligation to be part of the study and could withdraw from the study at any point without repercussions. A signed informed consent was obtained from the head of the medicine outlet, and verbal consent was from other staff of the participating medicine outlet.

### Setting

The study was conducted in the South West Region (SWR), one of ten regions in Cameroon with a population of over 1,384,286 persons as of January 2015 and population density of 32.96 persons/square kilometre [20]. The SWR has six administrative divisions each of which is further divided into subdivisions.

The Health System in Cameroon comprises three levels-the Central level develops health policies and strategies, and coordinates, regulates their implementation; the Intermediate level provides technical support to; the Peripheral (or operational) level which host the Health Districts and related services, implementing policies designed at the Central level There are eighteen health districts (HD) in which public, private and confessional (religious or faith-based) health facilities provide healthcare services. The public health facilities get their medicine supply from the regional drug programme (SWR Fund for Health Promotion) and sell following national guidelines, while private and confessional health facilities obtain theirs from the regional drug programme and from private suppliers. Access to medicines from both public and private facilities is at a cost.

### Selection of medicine outlets

A regional survey was conducted in all 4 sectors (public, private and confessional facility outlets, and private community pharmacies) in 6 survey areas (administrative divisions) and medicine outlets were selected using cluster and simple random sampling techniques, informed by the WHO/HAI methodology [21]. In each survey area, the main public health facility outlet was included and four other public facility outlets closest to the main public facility outlet randomly selected. Five private and five confessional outlets closest to the main public facility were also randomly selected. All community pharmacies within one kilometre of the main public facility medicine outlet were selected. Inclusion of the private community pharmacies was on the assumption that they offered the greatest possibility of getting medicines not found at the health facility outlets.

Of 108 medicine outlets selected, 63 medicine outlets were surveyed; 24 outlets of public facilities, 18 outlets of private facilities, 11 outlets of confessional facilities and 10 private

community pharmacies. Excluded from the study were health facilities without a pharmacy, non-consenting health facilities and private community pharmacies and medicine outlets requiring more than day's travel from the main public health facility. Some facility outlets which were initially selected were not eventually surveyed because during data collection, it was realised that they were remote settlements which could not be reached within a day, from the main public health facility and thus were excluded.

## Selection of medicines

Twenty-six EM were selected from the 19[th] WHO model core list [22]. These included: amiloride, amlodipine, aspirin, atenolol, atorvastatin, bisoprolol, captopril, clopidogrel, digoxin, enalapril, epinephrine, furosemide, glyceryl trinitrate, hydralazine, hydrochlorothiazide, isosorbide dinitrate, long acting penicillin, methyl dopa, nifedipine, propanolol, simvastatin, spironolactone and verapamil. Evidence, including their cost-effectiveness supports the use of these medicines in reducing morbidity and mortality and thus widely recommended for secondary prevention [20, 23]. The selection of these medicines was also guided by the list of medicines selected by a WHO team and other investigators in previous studies [15, 18, 24]. For the purpose of comparing between the availability of non-communicable and communicable diseases medicines, medicines for HIV/AIDS and tuberculosis; tenofovir, lamivudine, emtricitabine, efavirenz, zidovudine, nevirapine, abacavir, ritonavir-boosted lopinavir and ritonavir-boosted atazanavir were also surveyed in public and some confessional facilities since these are solely distributed at those facilities.

## Data collection

Information on availability and cost was obtained using an interviewer-administered data collection tool adapted from the medicine price data collection form in the World Health Organisation/Health Action International (WHO/HAI) Manual [25]. For each medicine outlet, availability was ascertained by seeing a given essential medicine on the day of the survey. Information collected for each medicine included the availability of the cheapest brand of any medicine dosage, cost per tablet and cost per pack or vial paid by the consumers. A pilot study based on a convenient sample was conducted to test the data collection tool for any limitations and appropriate adjustments made prior to conducting the main study.

## Data analysis

Data were analysed using SPSS® statistical software v.20 for Windows® (IBM, Chicago, USA). Availability was assessed by estimating the percentage of outlets in which any dose was found overall, by sector, and by setting (urban, semi-urban or rural). Availability was categorised as very low (< 30%), low (30–49%), fairly high (50–80%) and high (> 80%) [15]. For cost of medicines, the unit price (price per tablet or vial) was used to determine the price of each medicine. The median price (MP) of each medicine was also calculated in CFA/francs, and converted to US dollars using the exchange rate on the first day of data collection (1 US dollar = 602.46 francs CFA, February 2016). Median values were used as a better representation of the mid-point values as price is often skewed. The International Reference Price (IRP) was obtained from the 2014 edition of the International Drug Price Indicator Guide published by the Management Sciences for Health (MSH) [25]. The MSH International Drug Price Indicator Guide prices represent median prices of medicines supplied to LMIC by different suppliers and are recommended as the most useful standard for comparison [15]. Buyer Median Price (BMP) was used when Supplier Median Price (SMP) was unavailable. The median price ratio (MPR) was calculated for each medicine presentation. This ratio compares how much greater

or less the local MP is than the IRP [15, 25]. The MPR is an assessment of the purchasing efficiency of the system. A MPR of 1.5 or less was considered reasonable for all medicine forms [15]. To assess variability of prices across sectors, only medicines available in more than one sector and in more than one outlet in that sector were selected. Affordability was estimated using local MP, defined daily dose and average monthly wage of the lowest paid unskilled government worker, according to the WHO/HAI standard manual [15, 26, 27], which was 36,270 francs CFA, at the time of the survey [28]. The number of days' wages required to pay for one month course of treatment was also calculated. A month's course of chronic treatment requiring 1 day's wages or less was considered affordable [26, 27].

## Availability of data and material

Data cannot be made publicly available as the consent form did not include provisions for data sharing. Facility-level data can be made available to interested researchers with the approval of the Cameroon National Ethics Committee. The contacts of the Cameroon National Ethics Committee are available at https://healthresearchweb.org/en/cameroon/ethics_2120

## Results

### Distribution of outlets

The general characteristics of the health facilities with medicine outlets included in this study are presented in Table 1. Twenty-four (38.1%) public facility outlets, 18 (28.6%) private facility outlets, 11 (17.5%) confessional facility outlets and 10 (15.9%) private community outlets were surveyed. The catchment population of the surveyed facilities ranged from 1444 to 109000 individual, while the number of staff member per facility ranged from 4 to 225.

### Availability of medicines

For all medicines, the mean availability was 33% overall, 49.2% in private community, 44.0% in confessional facility outlets, 26.1% in private and 25.3% public facility outlets. Long acting penicillin had a high availability at 92.1%, and >80% in all sectors and categories; while glyceryl trinitrate and isosorbide dinitrate were completely absent. Hydrochlorothiazide was more than 70% available in all sectors and categories. Spironolactone and verapamil were only found in confessional and private community outlets. Atorvastatin, amiloride, bisoprolol, clopidogrel and simvastatin were only found in private community outlets. Neither aspirin 100mg nor 81mg was found in public facility outlets. The percentage availability within outlets is shown in Table 2. Anti-retroviral and anti-tuberculosis medicines were 100% available at all designated treatment centres.

**Table 1. Average population and personnel profile.**

| | Minimum | Maximum | Median |
|---|---|---|---|
| Average population served by the Health Facility in which outlet is located | 1444 | 109000 | 13269.50 |
| Number of Health workers in the Health Facility | 2 | 137 | 14.00 |
| Number of Nurses in the Health Facility | 1 | 69 | 10.00 |
| Number of Doctors (GPs) in the Health Facility | 0 | 13 | 1.00 |
| Number of Specialist in the Health Facility | 0 | 8 | 0 |
| Number of Internists or Cardiologists | 0 | 2 | 0 |
| Total number of Staff in the Health Facility | 4 | 225 | 22.00 |

**Table 2. Availability of all medicines across surveyed medicine outlets and by urbanicity.**

| Medicine | Percentage Availability | | | | | | |
|---|---|---|---|---|---|---|---|
| | Public facility outlet; % (n) | Confessional facility outlet; % (n) | Private facility outlet; % (n) | Private community outlet; % (n) | Urban; % (n) | Semi- urban; % (n) | Rural; % (n) |
| **Medicines for IHD/Stroke** | | | | | | | |
| Aspirin | 41.7 (10) | 81.8 (9) | 61.1 (11) | 100.0 (10) | 82.4 (14) | 64.0 (16) | 47.6 (10) |
| Atorvastatin | 0.0 (0) | 0.0 (0) | 0.0 (0) | 20.0 (2) | 5.9 (1) | 4.0 (1) | 0.0 (0) |
| Clopidogrel | 0.0 (0) | 0.0 (0) | 0.0 (0) | 30.0 (3) | 11 .8 (2) | 4.0 (1) | 0.0 (0) |
| Glyceryl trinitrate | 0.0 (0) | 0.0 (0) | 0.0 (0) | 0.0 (0) | 0.0 (0) | 0.0 (0) | 0.0 (0) |
| Isosorbide dinitrate | 0.0 (0) | 0.0 (0) | 0.0 (0) | 0.0 (0) | 0.0 (0) | 0.0 (0) | 0.0 (0) |
| Simvastatin | 0.0 (0) | 0.0 (0) | 0.0 (0) | 50.0 (5) | 17.6 (3) | 8.0 (2) | 0.0 (0) |
| **Medicines for heart failure** | | | | | | | |
| Digoxin | 4.2 (1) | 72.7 (8) | 16.7 (3) | 30.0 (3) | 35.3 (6) | 25.0 (6) | 14.3 (3) |
| Furosemide | 70.8 (17) | 81.8 (9) | 77.8 (14) | 100.0 (10) | 88.2 (15) | 80.0 (20) | 71.4 (15) |
| Spironolactone | 0.0 (0) | 27.3 (3) | 0.0 (0) | 70.0 (7) | 23.5 (4) | 20 .0 (5) | 4.8 (1) |
| **Medicines for Cardiac arrest** | | | | | | | |
| Epinephrine | 16.7 (4) | 90.9 (10) | 44.4 (8) | 0.0 (0) | 17.6 (3) | 40.0 (10) | 42.9 (9) |
| **Medicines for diabetes** | | | | | | | |
| Glibenclamide | 79.2 (19) | 100.0 (11) | 56.6 (10) | 100.0 (10) | 82.4 (14) | 84.0 (21) | 71.4 (15) |
| Insulin | 79.2 (19) | 81.8 (9) | 50.0 (9) | 100.0 (10) | 91.4 (16) | 72.0 (18) | 61.9 (13) |
| Metformin | 83.3 (20) | 90.9 (10) | 77.8 (14) | 100.0 (10) | 88.2 (15) | 92.0 (23) | 76.2 (16) |
| **Medicines for hypertension and other cardiac diseases** | | | | | | | |
| Amiloride | 0.0 (0) | 0.0 (0) | 0.0 (0) | 40.0 (4) | 23 (4) | 0.0 (0) | 0.0 (0) |
| Amlodipine | 16.7 (4) | 36.4 (4) | 5.6 (1) | 90.0 (9) | 47.1 (8) | 32.0 (8) | 9.5 (2) |
| Atenolol | 0.0 (0) | 9.1 (1) | 5.6 (1) | 40.0 (4) | 17.6 (3) | 12.0 (3) | 0.0 (0) |
| Bisoprolol | 0.0 (0) | 0.0 (0) | 0.0 (0) | 40.0 (4) | 11.8 (2) | 8.0 (2) | 0.0 (0) |
| Captopril | 0.0 (0) | 81.8 (9) | 5.6 (1) | 50.0 (5) | 29 .4 (5) | 28.0 (7) | 14.3 (3) |
| Enalapril | 0.0 (0) | 0.0 (0) | 5.6 (1) | 50.0 (5) | 23.5 (4) | 8.0 (2) | 0.0 (0) |
| Hydrochlorothiazide | 79.2 (19) | 90.9 (10) | 72.2 (13) | 100.0 (0) | 88.2 (15) | 84.0 (21) | 76.2 (16) |
| Long-acting penicillin | 95.8 (23) | 90.0 (10) | 83.3 (15) | 100.0 (10) | 100 (17) | 92.0 (23) | 85.7 (18) |
| Hydralazine | 0.0 (0) | 18.2 (2) | 0.0 (0) | 0.0 (0) | 0.0 (0) | 4.0 (1) | 4.8 (1) |
| Methyl dopa | 41.7 (10) | 63.6 (7) | 16.7 (3) | 50.0 (5) | 35.3 (6) | 32.0 (8) | 52.4 (11) |
| Nifedipine | 70.8 (17) | 63.6 (7) | 66.7 (12) | 0.0 (0) | 29.4 (5) | 68.0 (17) | 66.7 (14) |
| Propranolol | 12.5 (3) | 54.5 (6) | 11.1 (2) | 0.0 (0) | 11.8 (2) | 20.0 (5) | 19.0 (4) |
| Verapamil | 0.0 (0) | 9.1 (1) | 0.0 (0) | 20.0 (2) | 5.9 (1) | 8.0 (2) | 0.0 (0) |
| **Medicines for communicable diseases** | | | | | | | |
| Anti-retrovirals | 100 (8) | 100 (4) | NA | NA | 100 (3) | 100 (6) | 100 (4) |
| Anti-TB medicines | 100 (13) | 100 (4) | NA | NA | 100 (3) | 100 (7) | 100 (3) |

## Median prices, median price ratio and affordability of medicines

Among the surveyed medicines, only hydralazine (1.07), methyl-dopa (0.55) had reasonable MPR (MPR ≤1.5). Digoxin 0.25mg, furosemide 40mg, hydrochlorothiazide 25mg and nifedipine 20mg were affordable (cost a day's wage or less, of the lowest paid unskilled government worker, for 30 days of chronic treatment). Angiotensin converting enzyme inhibitors and beta blockers required 2 to 5 days' wages, while statins required 6 to 13 days' wages. Table 3 shows the cost and affordability of cardiovascular medicines. Medicines for HIV/AIDS and tuberculosis are distributed free of charge.

**Table 3. Median price, median price ratio and affordability of cardiovascular medicines.**

| Medicine dose | Defined daily dose | Median Price | | IRP/USD | Median Price Ratio | Cost of 30 days' treatment | | Number of days wages for 30 days treatment |
|---|---|---|---|---|---|---|---|---|
| | | CFAF | USD | | | CFAF | USD | |
| Aspirin 100mg | 100mg | 43.33 | 0.0719 | 0.0020 | 35.95 | 1300 | 2.16 | 1.08 |
| Amlodipine 5mg | 5mg | 116.83 | 0.1939 | 0.0252 | 7.69 | 3505 | 5.82 | 2.90 |
| Atenolol 50mg | 75mg | 107.00 | 0.1776 | 0.0103 | 17.24 | 4815 | 7.99 | 3.98 |
| Atorvastatin 20mg | 20mg | 279.33 | 0.4636 | 0.0552 | 8.40 | 8380 | 13.91 | 6.93 |
| Bisoprolol 5mg | 10mg | 116.67 | 0.1937 | 0.0660 | 2.93 | 7000 | 11.62 | 5.79 |
| Captopril 25mg | 50mg | 50.00 | 0.0830 | 0.0139 | 5.97 | 3000 | 4.98 | 2.48 |
| Clopidogrel 75mg | 75mg | 530.00 | 0.8797 | 0.0775 | 11.35 | 15900 | 26.39 | 13.15 |
| Digoxin 0.25mg | 0.25mg | 30.00 | 0.0498 | 0.0121 | 4.16 | 900 | 1.49 | 0.74 |
| Enalapril 10mg | 10mg | 210.00 | 0.3486 | 0.0059 | 59.08 | 6300 | 10.46 | 5.21 |
| Epinephrine 1mg | 0.5mg | 500.00 | 0.8299 | 0.3339 | 2.49 | 7500 | 12.45 | 6.20 |
| Furosemide 40mg | 40mg | 20.00 | 0.0332 | 0.0067 | 4.95 | 600 | 1.00 | 4.95 |
| Hydralazine 20mg | 100mg | 3000.00 | 4.9796 | 4.6717 | 1.07 | 450000 | 746.94 | 372.22 |
| Hydrochlorothiazide 25mg | 25mg | 39.00 | 0.0647 | 0.0043 | 15.05 | 1170 | 1.94 | 0.97 |
| Methyl dopa 500mg | 1000mg | 40.00 | 0.0664 | 0.1200 | 0.55 | 2400 | 3.98 | 1.99 |
| Nifedipine 20mg | 30mg | 27.50 | 0.0456 | 0.0250 | 1.82 | 1238 | 2.05 | 1.02 |
| Propranolol 40mg | 160mg | 30.00 | 0.0498 | 0.0075 | 6.64 | 3600 | 5.98 | 2.98 |
| Simvastatin 20mg | 30mg | 360.71 | 0.5987 | 0.0531 | 11.27 | 16232 | 26.94 | 13.43 |
| Spironolactone 25mg | 75mg | 120.00 | 0.1992 | 0.0432 | 4.61 | 10800 | 17.93 | 8.93 |
| Verapamil 240mg | 240 | 330.00 | 0.5478 | 0.0879 | 6.23 | 9900 | 16.43 | 8.19 |

IRP: International reference price

## Variability in cost and affordability of medicines

The cost of medicines was highest in private community outlets, relatively similar between confessional and private facility outlets and lowest in public facility outlets. The variability in prices and affordability across sectors for some selected medicines is shown in Tables 4 and 5.

## Discussion

In this survey exploring the availability and affordability of essential CVD medicines, we found mean availability of 33%, ranging between 25.3% in public facility outlets and 49.2% in private community pharmacies, in the South West Region of Cameroon. This was in sharp contrast with 100% availability and HIV and anti-tuberculosis medicines at approved

**Table 4. Median prices across sectors for some selected medicines.**

| Medicines | Median Prices (CFAF) per Sector | | | |
|---|---|---|---|---|
| | Public facility outlet | Confessional facility outlet | Private facility outlet | Private community outlet |
| Aspirin 100mg | NA | 12.5 | 10.0 | 45.5 |
| Captopril 25mg | NA | 50.0 | NA | 94.3 |
| Digoxin | NA | 30.0 | 50.0 | 91.7 |
| Epinephrine 1mg/ml | 80.0 | 500.0 | 500.0 | NA |
| Furosemide 40mg | 5.0 | 25.0 | 27.5 | 76.6 |
| Hydrochlorothiazide 50mg | 5.0 | 50.0 | 25.0 | NA |
| Methyl dopa 250mg | 25.0 | 50.0 | 40.0 | 133.3 |
| Nifedipine 20mg | 20.0 | 30.0 | 37.5 | NA |
| Spironolactone | NA | 50.0 | NA | 120.0 |

**Table 5. Affordability of some selected medicines across sectors.**

| Medicine | Defined daily dose | Public facility outlet | | Confessional facility outlet | | Private facility outlet | | Private community outlet | |
|---|---|---|---|---|---|---|---|---|---|
| | | Cost of 30 days treatment (CFAF) | Number of day's wages for 30 days | Cost of 30 days treatment (CFAF) | Number of day's wages for 30 days | Cost of 30 days treatment (CFAF) | Number of day's wages for 30 days | Cost of 30 days treatment (CFAF) | Number of day's wages for 30 days |
| Furosemide 40mg | 40mg | 150 | 0.1 | 750 | 0.6 | 825 | 0.7 | 2298 | 1.9 |
| Hydrochlorothiazide 50mg | 25mg | 75 | 0.06 | 750 | 0.6 | 375 | 0.3 | 1170 | 1 |
| Methyl dopa 250mg | 1000mg | 3000 | 2.5 | 6000 | 5 | 4800 | 4 | 15996 | 13.2 |
| Nifedipine 20mg | 30mg | 900 | 0.7 | 1350 | 1.1 | 1687.5 | 1.4 | NA | NA |

treatment facilities. Medicines cost highest in private community pharmacies, were similar between confessional and private facility outlets and lowest in public facility outlets. Digoxin 0.25mg, furosemide 40mg, hydrochlorothiazide 25mg and nifedipine 20mg were the only four affordable out of the 26 surveyed medicines.

Overall, the availability of hydrochlorothiazide in all outlets was 82.5%, consistent with 85.1% in Brazil, 100% in Sri Lanka but much higher than 5.9% in Bangladesh in 2007 [27], and the 43.7% reported in Haiti in 2013 [29]. Hydrochlorothiazide 50mg availability across all sites was higher than the 40% previously reported in the West region of Cameroon [15]. However, Atenolol 50mg availability in our study was consistent with previous report from the West region of Cameroon, but much lower than 66% in Sri Lanka in 2014 [24] and 44.5% in Haiti. Availability of furosemide (79.4%) and propranolol (17.5%) were lower than 100% and 20% reported in the West region of Cameroon. Furosemide availability in our study was higher than 56.2% in Haiti but lower than 95% reported but lower than in Sri Lanka. Simvastatin' availability was close to previous report in the West region of Cameroon, but higher than 2.3% in Haiti but considerably lower than 49% in Sri Lanka. The availability of oral antidiabetics-glibenclamide and metformin was higher in the rural areas than what was reported in the West region of Cameroon. Only the 500mg formulation of aspirin was available in public facilities probably because the low dose formulation is not found Cameroon's National Essential Medicine List.

Availability of medicines in urban and semi-urban outlets was considerably higher than in rural outlets, probably because rural outlets preferably stock medicines for communicable diseases [14] which are symptomatic at onset and readily diagnosed as opposed to cardiovascular diseases and diabetes which are asymptomatic until complications set in, at which point patients migrate to urban centres for further care. This variability in the availability of EM in outlets in different settings may be accounted for, at least in part, by the concentration of private medicine outlets (private facilities and community pharmacies) in the urban and semi-urban settings. The low availability in public sector outlets and rural outlets compared to other sectors may be due to insufficient spending to procure sufficient medicines for patients' needs. It may also be as a result of low demand, perhaps suggesting that the medicines which were not available are not being prescribed by health professionals. Further, it may portray a poor distribution system to more remote medicine outlets.

The cost (price) of medicines was highest in private community outlets, relatively similar between confessional and private facility outlets, but slightly higher in the latter. The lowest prices were found in public facility outlets. Median prices were several folds higher than the IRP, especially in the non-public sectors. This may be due to the procurement system being inefficient, absence of price control or higher mark-ups in the private outlets to cover running cost and make profit. Aspirin in the private sector (private health facilities and private

community pharmacies) cost more than 35 times the IRP (MPR = 35.96) and hydrochlorothiazide more than 15 times the IRP (MPR = 15.06). This is higher than in Nepal and Sri Lanka where the MPRs for aspirin of 12 and 10 respectively were reported. The MPR of hydrochlorothiazide was reported to be 11 in Sri Lanka [18].

Only digoxin, furosemide, hydrochlorothiazide and nifedipine were affordable (cost a day's wage or less for a course of one month treatment). The rest of the medicines were largely unaffordable with beta blockers and angiotensin converting enzyme inhibitors requiring more than 2 days' wages for a month-long treatment. Considering that some patients may be taking multidrug regimens, the relative unaffordability of single medicines surmises that multidrug regimens will be even more unaffordable We found that treating a patient with ischaemic heart disease with a combination of BB (atenolol), ACEI (captopril), a statin (simvastatin) and aspirin cost 19.8 days' wages for a month of chronic treatment, which is greater than in Sri Lanka (1.5 days' wages), Nepal and Pakistan (5 days' wages) and Malawi (18.4 days' wages) [15, 27], but lower than 41.3 days' wages reported in the West region of Cameroon [27]. In the public sector where cost was relatively lower, the availability of medicines was low. As such, patients either turn to confessional and private sectors where medicines are more available albeit the comparatively higher cost, or forgo treatment altogether [13]. The unaffordability of medicines has been blamed largely on poverty, lack of effective insurance systems and out-of-pocket payments [14, 15, 29]. The concept of insurance systems is relatively new in Cameroon [6]. A study reported that only 4.4% of Cameroonian informal sector workers had health insurance [9]. As such, chronic treatment stretches household resources to the limit, often competing with other basic household needs.

These notwithstanding, the availability of medicines for communicable diseases; anti-TB and anti-retroviral medicines was 100% in all designated treatment centres irrespective of sector or urban-rural location. This most likely implies that very little attention is directed towards CVD compared to HIV/AIDS, tuberculosis and other communicable diseases. Indeed, a National Multi-sectoral Action Plan on NCDs is yet to be made available publicly. This is rather unfortunate because, even though communicable diseases are undeniably more prevalent, the prevalence of CVD is steadily rising due to the worsening risk profiles, which are projected to almost double by 2030 if current trends persist [27]. The findings in this study suggest that management of CVD and its risk factors remains a major public health challenge as treatment is observed cost several days' wages of the lowest-paid unskilled government worker. Furthermore, medicines are largely purchased by out-of-pocket payments due to the absence of universal health coverage and few health insurance schemes. Studies show that many people fall into the "medical poverty trap" every year due to CVD-related expenditures [30].

Going by these findings, it is therefore evident that we are a long way from achieving the WHO global targets for the control of NCD, especially global target 9. Since the distribution of medicines for communicable diseases is guided by national policy, it is therefore imperative that medicines for CVD fall under similar policies. It is important to appraise the situation of CVD, in terms of availability of diagnostic equipment, evidence-based management on a nation-wide scale, and design policies which aim at addressing the shortcomings, with the ultimate goal of attaining the WHO global targets but even more importantly, providing low-cost medicines on a wide scale to be used for primary and secondary prevention of cardiovascular events. Also, given that the WHO is currently focusing on improvement in access to affordable EM, it is important to have reliable mechanisms which can monitor trends in the supply system availability over time because relying on ad hoc systems may lead to information gaps and wastage of limited resources [14].

This is the first region-wide survey on this scale in Cameroon using methods adapted from the WHO/HAI methodology which has been validated in previous studies. This survey has provided a situational analysis of cardiovascular diseases in the South West region, in terms of the availability and affordability of essential medicines. While it may be impractical to extrapolate findings in this study to the entire country, the findings however, provide a glimpse at what may obtain in the other nine regions.

Our study has a number of limitations: availability was based on the presence of a particular medicine in the medicine outlet at the time of the survey. It is possible that outlets may have just run out of medicines at the time of the survey and this may not be true a reflection of their availability. Secondly, drug stores and roadside medicine vendors, which probably play an important role in the distribution of medicines, were not included in the survey. However, patients are usually advised to purchase their medicines from hospitals and private community pharmacies, to avoid use of substandard products. The method of sampling which followed the WHO/HAI standard manual may also have an effect on the results.

## Conclusion

Overall, the mean availability of essential medicines for CVD was 33%, much lower than the 80% recommended by the WHO Global Action Plan for Prevention of NCD. Additionally, the available medicines were largely unaffordable. While recognizing the key impact of primary prevention, there is an urgent need for the health system to scale up availability and affordability of medicines for treatment and secondary prevention of cardiovascular disease. Lessons from prior population level efforts in HIV and TB treatment and prevention are likely to offer guidance for setting implementation agendas in the sphere of CVD.

## Supporting information

**S1 File. Finalised data collection tool.**
(DOCX)

## Acknowledgments

Immense gratitude goes to all our respondents.

## Author Contributions

**Conceptualization:** Anastase Dzudie, Epie Njume.

**Data curation:** Epie Njume.

**Formal analysis:** Epie Njume.

**Methodology:** Epie Njume, Martin Abanda.

**Project administration:** Anastase Dzudie, Epie Njume.

**Resources:** Epie Njume.

**Supervision:** Anastase Dzudie, Leopold Aminde, Marie Solange Doualla, Marcelin Ngowe, Andre P. Kengne.

**Writing – original draft:** Epie Njume.

**Writing – review & editing:** Anastase Dzudie, Epie Njume, Martin Abanda, Leopold Aminde, Ba Hamadou, Bonaventure Dzekem, Marcel Azabji, Marie Solange Doualla, Marcelin Ngowe, Andre P. Kengne.

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
