## [Decision Letter · Decision Letter 0]

27 Nov 2019

PONE-D-19-21246

Availability, Cost and Affordability of Essential Cardiovascular Disease Medicines in the South West Region of Cameroon: Preliminary Findings from the Cameroon Science for Disease study

PLOS ONE

Dear Dr. Njume,

Thank you for submitting your manuscript to PLOS ONE. After careful consideration, we feel that it has merit but does not fully meet PLOS ONE’s publication criteria as it currently stands. Therefore, we invite you to submit a revised version of the manuscript that addresses the points raised during the review process.

This is a well written Manuscript. Address the minor comments raised by the reviewers and return for re-consideration.

We would appreciate receiving your revised manuscript by 15th December 2019. To enhance the reproducibility of your results, we recommend that if applicable you deposit your laboratory protocols in protocols.io, where a protocol can be assigned its own identifier (DOI) such that it can be cited independently in the future. For instructions see: http://journals.plos.org/plosone/s/submission-guidelines#loc-laboratory-protocols

We look forward to receiving your revised manuscript.

Kind regards,

Geofrey Musinguzi, MPH, PhD

Academic Editor

PLOS ONE

Journal Requirements:

2. Please provide a copy of the finalised data collection tool used in the study, in both the original language and English, as Supporting Information.

3. Please ensure that your references are numbered in the order they appear in the text. For example, in the Introduction section the reference numbers appear to go from 18 straight to 22.

4. Please cite and discuss the following related work in the body of the manuscript (introduction and discussion): https://journals.plos.org/plosone/article?id=10.1371/journal.pone.0111812

Reviewers' comments:

Reviewer's Responses to Questions

**Comments to the Author**

1. Is the manuscript technically sound, and do the data support the conclusions?

Reviewer #1: Yes

Reviewer #2: Yes

2. Has the statistical analysis been performed appropriately and rigorously? 

Reviewer #1: Yes

Reviewer #2: Yes

3. Have the authors made all data underlying the findings in their manuscript fully available?

Reviewer #1: Yes

Reviewer #2: Yes

4. Is the manuscript presented in an intelligible fashion and written in standard English?

Reviewer #1: Yes

Reviewer #2: Yes

5. Review Comments to the Author

Reviewer #1: 1. Selection of essential drugs list needs to be revisited. Essentialness of isosorbide mono nitrate and methyldopa is controversial

2. The drug list (127) does not mention about anti-diabetic drugs in the table 2

3. Definition of affordability needs to be defined at the beginning and the cutoff taken may need revision as it is too low

Reviewer #2: Comments are available on attachment

6. PLOS authors have the option to publish the peer review history of their article (what does this mean?). If published, this will include your full peer review and any attached files.

Reviewer #1: Yes: Chamila Mettananda

Reviewer #2: No

---

## [Author Response · Author response to Decision Letter 0]

13 Jan 2020

Availability, Cost and Affordability of Essential Cardiovascular Disease Medicines in the South West Region of Cameroon: Preliminary Findings from the Cameroon Science for Disease study

Dear Editor,

Season’s greetings.

Below is a point-by-point response to each of the points raised by the you, and the reviewer(s):

Introduction;

Line 47. CVD is the leading cause of death globally. ……

Line 49/50. Recent projections show that suggest that deaths from CVDs will continue to rise, surpassing all other causes by 2030.

I thought these two lines from the start are contradictory. Because if the first says CVD is the leading cause of death globally, and then the next says it will surpass all other causes by 2030, then it needs to be clarified.

Response

Thank you for this pertinent comment. The lines have been clarified to read:

Line 47/48/50: Globally, cardiovascular disease (CVD) is the leading cause of death with 80% of these deaths occurring in low- and middle-income countries (LMIC). 

Methodology

Line 99. I think it would be important to throw in some line that explains what a health district (HD) and how it is structured, because this concept is used in other countries but it actually mean varied things. OR even probably explain aliitle elaborately how the Cameroon health system is structured and the medicine distribution and supply chain. Do people access medicines from public and private facilities at a cost? 

Response

The Health System in Cameroon is explained, and the question on access to medicines from public and private facilities answered:

Lines 102-105/110: The Health System in Cameroon comprises three levels-the Central level develops health policies and strategies, and coordinates, regulates their implementation; the Intermediate level provides technical support to; the Peripheral (or operational) level which host the Health Districts and related services, implementing policies designed at the Central level

Access to medicines from both public and private facilities is at a cost

Line 108/109. What was the justification for choosing the main public hospital, AND then four others closest to it? I think a justification line needs to be added in there, because I note that in line 112/113, the community Pharmacies closest are included and their inclusion was justified. But I am wondering what that also implies for introducing clustering biases an effects in the data.

Response

The process of selection of medicine outlets is in line with the WHO/HAI standard methodology (page 27). Community pharmacies were not mentioned in the WHO/HAI manual hence their inclusion was justified. This is in the manuscript; (line 114/115)

Line 118, implies to me that the sampling and selection of outlets especially community pharmacies, was based more on convenient sampling probably to reduce study costs. But I am wondering whether this could have had recognizable effects on the data an ultimately the understanding and interpretation of results?

Response 

We thank the reviewer for this comment. We agree with the reviewer that our sampling procedure could have effected of the on the results. We have acknowledged this as a limitation in the discussion (limitation section). 

Line 171/174. I thought this section touches on ethics and data confidentiality issues and therefore could have fitted well in the section of ethics at the start of the methodology section. But again this depends on how PLOS wants a paper structured.

Response 

This section is not strictly related to ethical and data confidentiality of the study methodology but rather, a response to PLOS requirement of make data publicly available in all cases, with few exceptions.

Results

Line 176/181. I thought the write-up in this section was sufficient and in any case could be expanded, making the Table 1, irrelevant because its information has already been explained. I find it not adding any value to this.

Response 

The table was added instead of expanding on the write-up for clarity It provides additional information on the cadre of personnel at the facilities hosting outlets, which can be an indication of the services rendered, and thus the kinds of medications to be found at these outlets. 

Line 195 onwards. I find that the write-ups are virtually a repetition of the table 2. It is always recommended that once you use a table, you could only explain the key issues in the table that the readers should take note of. But you cannot repeat an entire table by writing.

Response

Sections of text which were virtually repetitive from 211 following. 

Line 236. I should have made this comment in methodology. The study considered the wage of a lowest paid unskilled government worker. Can they include the justification for using this reference category? Is it that the lowest paid government workers are the ones who have a higher burden of disease in this CVD or NCD area? Or is it that once the lowest paid Government worker can afford this treatment then everyone else above them is expected to afford? How does a lowest paid government worker compare with a peasant individual without any gainful work or in agriculture? Would it have been useful to compare with incomes of those in agriculture or no gainful work at all? I think some clarification for selection of the reference case may be convincingly necessary.

Response 

We agree with the reviewer that other references could have been used. However, we preferred to use an internationally recommended reference. The lowest paid unskilled government worker is used as the reference in the WHO/HAI standard manual. We have added this in the corresponding section (line 173/174).

Line 280. I found Table 4, not necessary. In the first place I don’t know what the figures are, and whether they are dollar values or something else or percentages. I would recommend that Table 5 communicates better the idea of cost and affordability. This is because cost, moving forward is quoted in dosage (30 days of required dosage). So I would think Table 4 is generally irrelevant.

Response

Table 4 compares the median prices in CFA (seen at the top of the table) for a daily dose of across medicine outlets.

General 

**These drugs seem to have been taken as a package. However, I was wondering for CVD for example, whether an entire drug package should be available to treat a CVD case OR there is this one or two essential drug that if not available, then a patient is in problems. Could you have identified that “essential” of “essentials” and assessed their availability? 

Response

The medicines were selected from the “WHO Model List of Essential Medicines. 19th List. 2015” cited (24) in the Methodology section. CVDs typically require more than one medicine and as the condition, worsens or progress, or other complications set in, even more medicines may need to be added.

Between lines 321/324, we reported and compared affordability of a multidrug regimen for the treatment of ischaemic heart disease. 

*****I would have expected some result on the possible reasons or factors associated with the variation in MPs across the sectors, and this would tie in well with an earlier comment I made of …describing the structure of the health system and probably the supply chain so we can know where the costs and therefore higher markups are likely to be associated. 

Response

The Health System in Cameroon has been described in a response to made above.

Possible factors responsible for variation in MPs are found on lines 318/319

***We could have also benefitted from knowing whether geographical considerations have an influence on costs of drugs, that’s looking at urban and rural, those within vicinities of big public facilities compared to those a little further away from them. How do their medicine prices compare? I know you did this comparison only on ‘availability’. 

Response

Availability of medicines was lowest in public sectors which generally had the lowest MPs (for medicines which were available) and were more likely to be located in rural settings. Further, the prices in these public outlets were the same, irrespective of the geographical location. Private facility and community outlets which had highest level of availability and MPs are mostly based in the semi-urban and urban settings (lines 312). We could not assess if geographical locations influenced the affordability because the medicines were not available to begin with. 

Discussion

**Why was aspirin missing in virtually public facilities? Does it have something to do with essential medicines list? Or change in prescriptive policy? I thought I would find this argument in the discussion.

Response

Thank you. Line 307/308 addresses this remark. “Only the 500mg formulation of aspirin was available in public facilities probably because the low dose formulation is not found Cameroon’s National Essential Medicine List”. 

Line 309/315. I would have thought some small amount of qualitative would have explained these facts you are imputing on doctors not prescribing certain drugs. I think a KI interview would have added value. 

Response

Unfortunately, this was not assessed. We acknowledge the pertinence and will be exploring this as a follow-up of this study

Line 352. I think the issue of a reference wage case, needs to be explained, otherwise it may end-up in this section of limitations. But it should be elaborated in the earlier sections. 

Response

We used an internationally recommended reference, the lowest paid unskilled government worker which is recommended in the WHO/HAI standard manual. We have explained this in the corresponding section (line 173/174).. 

**** I thought results would have also been contextualized alittle more. Availability was very low in public facilities. How does this relate with the broader context of the Cameroon economy, public expenditure on health, per capital health expenditure, or generally the health financing structure of Cameroon, or the generic problems that are often reported of medicine distribution and supply chain challenges, pilferage (medicine thefts from public facilities ending into private), the big numbers that report in public facilities and hence depleting the available stocks, medicine waste, etc. These issues are not unique, and so I am surprised that they seem not to feature prominently in the discussion.

Response

This context is provided in lines 314/316

A brief context of the Health System and Drug Procurement and distribution is presented in lines 102/110 in the methodology section.

***I did ask a question, whether medicines in public facilities were also paid for. And this comment could be because you never explained to us the structure of Cameroon health system. I know many public health systems are free, and so I could not understand what cost mean for a public system that may be free being costed. 

Response

I have indicated in the methodology section that medicines are paid for in public facilities., line 110

***Some comments about if there exists a policy reform now on NCDs in Cameroon may be good to understand whether NCD agenda has now been put on the priority list of issues. Some health systems are still focused on CDs and even their EMS has not changed to reflect the new NCD realities. Could this have an impact on the low stocks we have witnessed in the public and the confessional sectors? 

Response

A National Multi-sectoral Action Plan on NCDs was elaborated by a working group (I represented my organisation) towards the end of 2018 but the final document has never been made available publicly.

Indeed, more focus rests on CDs clearly reflected in the 100% availability and cost-free medications for HIV/AIDS and TB. This has been added to the discussion section (340/341).

---

## [Editor Report · Decision Letter 1]

4 Feb 2020

Availability, Cost and Affordability of Essential Cardiovascular Disease Medicines in the South West Region of Cameroon: Preliminary Findings from the Cameroon Science for Disease study

PONE-D-19-21246R1

Dear Dr. Mesumbe,

We are pleased to inform you that your manuscript has been judged scientifically suitable for publication and will be formally accepted for publication once it complies with all outstanding technical requirements.

With kind regards,

Geofrey Musinguzi, MPH, PhD

Academic Editor

PLOS ONE
---

## [Editor Report · Acceptance letter]

25 Feb 2020

PONE-D-19-21246R1 

Availability, Cost and Affordability of Essential Cardiovascular Disease Medicines in the South West Region of Cameroon: Preliminary Findings from the Cameroon Science for Disease study. 

Dear Dr. Njume:

I am pleased to inform you that your manuscript has been deemed suitable for publication in PLOS ONE. Congratulations! Your manuscript is now with our production department. 

With kind regards,

on behalf of

Dr. Geofrey Musinguzi 

Academic Editor

PLOS ONE